# Stabilizing Metallic Na Anodes via Sodiophilicity Regulation: A Review

**DOI:** 10.3390/ma15134636

**Published:** 2022-07-01

**Authors:** Chenbo Yuan, Rui Li, Xiaowen Zhan, Vincent L. Sprenkle, Guosheng Li

**Affiliations:** 1School of Materials Science and Engineering, School of Chemistry and Chemical Engineering, Anhui University, Hefei 230601, China; c20201013@stu.ahu.edu.cn (C.Y.); c20201073@stu.ahu.edu.cn (R.L.); 2Battery Materials and Systems Group, Pacific Northwest National Laboratory, Richland, WA 99352, USA; vincent.sprenkle@pnnl.gov

**Keywords:** sodium-metal batteries, sodium-metal anodes, Na wetting, interfacial contact, dendrite growth

## Abstract

This review focuses on the Na wetting challenges and relevant strategies regarding stabilizing sodium-metal anodes in sodium-metal batteries (SMBs). The Na anode is the essential component of three key energy storage systems, including molten SMBs (i.e., intermediate-temperature Na-S and ZEBRA batteries), all-solid-state SMBs, and conventional SMBs using liquid electrolytes. We begin with a general description of issues encountered by different SMB systems and point out the common challenge in Na wetting. We detail the emerging strategies of improving Na wettability and stabilizing Na metal anodes for the three types of batteries, with the emphasis on discussing various types of tactics developed for SMBs using liquid electrolytes. We conclude with a discussion of the overlooked yet critical aspects (Na metal utilization, N/P ratio, critical current density, etc.) in the existing strategies for an individual battery system and propose promising areas (anolyte incorporation and catholyte modifications for lower-temperature molten SMBs, cell evaluation under practically relevant current density and areal capacity, etc.) that we believe to be the most urgent for further pursuit. Comprehensive investigations combining complementary post-mortem, in situ, and operando analyses to elucidate cell-level structure-performance relations are advocated.

## 1. Introduction

The overreliance of our society on fossil fuels for electricity generation has led to a severe energy crisis and environmental issues, motivating the development of reliable electrochemical energy storage systems to store renewable energy, namely wind and solar, for large-scale grid and vehicle electrification applications [1]. Currently, various battery technologies are being actively investigated, including lead-acid batteries, redox-flow batteries, Li/Na-ion batteries, high-temperature batteries utilizing molten Li/Na anodes, such as Li/Na-S or Li/Na metal-halide (Li/Na-MH) batteries. Among them, Na-metal batteries (SMBs) are of particular interest due to their high cell voltages, high theoretical specific energy densities, and the abundance of Na and other components used in the batteries [2]. In review of the progress made in developing various SMBs, including molten, all-solid-state, and conventional liquid-electrolyte-based SMBs, it is generally recognized that all SMBs share a similar Na wetting challenge, although to different extents, and wetting-enhancing therapies, such as forming alloys, have been extended from one system to another.

Originally, SMBs were based on molten Na-S batteries or Na-MH batteries operating at around 300 °C with *β″*-Al_2_O_3_ solid electrolyte (BASE). While both types of molten SMBs are of technological interest and have been commercialized, Na-MH batteries, such as Na-NiCl_2_ batteries, are particularly promising due to their inherent safety in terms of cell failure relative to Na-S batteries, the less corrosive active materials compared to S and polysulfides, and easier cell assembly in the discharged state without handling hazardous Na metal and metal halides [3,4]. Unfortunately, the widespread adoption of both Na-S and Na-MH batteries in grid-level energy storage has been dwarfed by the safety concerns associated with their high operation temperature, plus the unstoppable emergence of Li-ion batteries. One of the main challenges encountered while reducing the operation temperature is the poor wettability of molten sodium on BASE and thus large interfacial resistance at lower temperatures. Recent successes in BASE surface modifications (i.e., via Pb and Bi) have resolved Na|BASE wetting issues to a large extent and greatly improved Na-ion transport at the interface, enabling intermediate- and even low-temperature operation of molten Na-MH and Na-S batteries [5,6,7,8].

In recent years, all-solid-state SMBs have received considerable interest due to their attractive safety from the use of nonflammable and thermally stable solid electrolytes (SE), and potentially high energy density from the use of Na metal anodes and high-voltage cathodes [2,9,10,11,12]. However, they suffer from the same or even more severe Na|SE wetting issues since they are supposed to run at room or ambient temperatures below the melting point of Na at 98 °C. The poor Na|SE contact will lead to large interfacial resistances and inhomogeneous Na stripping/plating morphologies, resulting in dendrite growth and eventually cell short circuits [13,14,15]. To tackle this issue, wetting-enhancement-based therapies have been extended from molten SMBs to all-solid-state SMBs.

The Na metal anode is the essential component for room-temperature sodium-metal batteries (SMBs), such as Na-S, Na-Se, and Na-O_2_. However, the practical application of liquid-electrolyte-based SMBs (thereafter referred to as conventional SMBs for the purpose of distinguishment) has been troubled by extensive volume change and mossy/dendritic growth during Na electrodeposition [16]. To address these issues, 3D hosts are widely considered since they not only provide abundant space to encapsulate Na metal but also homogenize the electrical field to inhibit the Na dendrite formation. However, achieving homogeneous and stable distribution of metallic over the host matrix is still challenging as most host materials are sodiophobic [17]. Introducing sodiophilic sites on the host has widely proven successful in enhancing the Na metal wettability, and, when followed by molten Na infusion, can be readily used to fabricate reliable Na metal anodes. The wetting-enhanced hosts can further reduce the Na nucleation energy and enable homogeneous Na plating/stripping. Such a design is thus able to minimize the Na dendrite formation, improve the cycling coulombic efficiency (CE), and achieve a long cycle life [16]. Moreover, the 3D host commonly exhibits good flexibility and controllable thickness, endowing the resultant Na composite with good machinability and excellent thickness tunability, beneficial for practical applications [18].

Clearly, the sodiophilicity (of the solid electrolytes or 2D/3D Na hosts) has been widely deemed vital for achieving stable Na metal anodes in either solid- or liquid-electrolyte-based SMBs. Thus far, there already exist quite a few comprehensive reviews involving Na-metal anodes and their interphases with liquid or solid electrolytes [2,9,16,17,19]. However, these existing reviews either discuss major advancements in all Na-based batteries [2,17,19] or only concern progress made in solid-state Na batteries [9] and dendrite issues [16]. To date, a focused review on a variety of strategies centered on sodiophilicity regulation is not yet available for research in SMBs. Therefore, the present review, for the first time, recognizes the universal Na wetting issues crossing molten, all-solid-state, and conventional liquid-electrolyte-based SMBs and provides an in-time summary of sodiophilicity regulation strategies and their corresponding electrochemical performances in different types of SMBs.

We will begin with briefly discussing these anode interface engineering strategies in Section 2. In Section 3, we then discuss recent reports on improving the Na anode wettability on two important SEs, BASE and NASICON-type Na_3_Zr_2_Si_2_PO_12_ (NZSP) solid electrolytes, assessed in all-solid-state SMBs. In Section 4, carbon, alloy, metal, and MXene-based hosts will be individually reviewed in detail, highlighting the role of wetting-improvement in stabilizing Na metal anode towards dendrite-free long-cycling SMBs using liquid electrolytes. We would like to note here that, although the three SMB systems are all based on Na metal anodes subjected to similar wetting challenges, they differ from each other considerably in terms of battery components (i.e., electrolyte and/or cathode), cell configuration, operating temperature, etc. Such differences limit the comparisons in the wetting strategy and the battery performance to an individual SMB system. In the last section, we conclude by pointing out some overlooked yet critical aspects in implementing the wetting-based strategies in an individual battery system and suggest promising directions for further pursuit. This review is expected to inspire new ideas and concepts in resolving critical metallic anode issues and broadening the applications of SMBs in emerging energy storage fields.

## 2. Molten Sodium-Metal Batteries

Among the Na-based batteries considered for large-scale grid storage applications, molten sodium-metal batteries have recently regained popularity mostly due to the recent advancements in the development of intermediate-temperature molten sodium-metal batteries (SMBs), including Na-metal halide (Na-MH) and Na-S batteries based on a *β″*-Al_2_O_3_ solid electrolyte (BASE) [3,4,20,21,22,23]. BASE electrolytes have a high ionic conductivity (∼1 and 250 mS cm^−1^ at 25 and 300 °C, respectively), an outstanding mechanical stability, and a wide electrochemical window. Unfortunately, they suffer from poor molten Na wetting, mandating the use of high temperatures (~300 °C) to ensure intimate Na|BASE interfacial contact and thus sufficient Na-ion conductivity [24]. It is of paramount significance to improve the molten Na wetting on BASE in order for intermediate-temperature molten SMBs to achieve further market penetration in grid applications.

Thus far, a great deal of promising and straightforward effort has been devoted to tackling this wetting challenge. Among the various strategies that have been proposed, the introduction of alloyable metallic coatings (e.g., Sn, Bi, In, Pb) or direct use of alloy anodes (i.e., Na-Cs alloy) on the BASE surface have proven effective [5,6,7,8,25,26]. For example, Lu et al. at PNNL reported a tactic of forming Na-K/Rb/Cs alloys that can markedly improve the wettability of liquid anodes on BASE [6]. As shown in Figure 1a, the Na-Cs alloy anode shows much higher wettability on BASE than that of pure Na at various temperatures, achieving a perfect wetting (contact angle near 0°) at 250 °C. In light of computational modelling of the wetting process, the droplets can be observed to be reduced in height while expanded in breadth as the Cs content in the alloys rises (Figure 1b–e). As a result, as depicted in Figure 1f,g, the Na-S cells with Na-Cs alloy anodes display only a slight capacity degradation when operated at 150 °C, delivering a capacity fade rate of <3% over 100 cycles. Even at a lower temperature (95 °C), the corresponding Na-Cs||S cells maintained a high capacity of 330 mAh g^−1^ under a constant current of 7 mA (~C/7). Later, Chang et al. from the same group successfully enhanced Na wettability on BASE via a micron-sized Pb decoration realized by treating lead acetate trihydrate at 400 °C (Figure 1h) [7]. Benefiting from the formation of the Na-Pb alloy, a promising Na wetting phenomenon signified by the “sunny-side-up” shape was observed when Na droplets were placed on the BASE surface, as shown in Figure 1i,j. Following this work, recently, Li et al. further improved the sodium wettability on BASE at 120 °C by raising the surface treatment temperature of lead acetate trihydrate and thus thinning the oxidation layer on Pb surface. The resulting low-temperature Na-S cell can reach a capacity as high as 520.2 mAh g^−1^ and stable cycling over 1000 cycles [20]. In an effort to reduce the toxic Pb amount, the same group developed a coating on BASE consisting of a porous carbon network decorated with PbO_x_ (0 ≤ x ≤ 2) nanoparticles [21]. An improved molten sodium wetting with a spreading temperature as low as 110 °C was achieved, along with ultrastable symmetric cell cycling for 6000 cycles at 120 °C. They found that sodiation could eventually break through the oxide shell and form metallic components at higher temperatures, as shown in Equation (1).
PbO, PbO_2_ + Na → Pb^0^ + Na-Pb + byproducts (Na_2_O)(1)

Furthermore, this modification drastically lowered the Pb content to approximately 6 wt.% in the anode interface, and a complete elimination of the toxic Pb was also demonstrated by replacing Pb with environmentally benign Sn [21]. Some other impressive coatings have also been successful, such as iron oxide [27], various carbon-based modified materials [28,29], and Pt or Ni meshes [30,31].

In addition to BASE, NASICON-type solid electrolytes based on Na_3_Zr_2_Si_2_PO_12_ (NZSP), first proposed by Goodenough and Hong et al. in 1976, have also been applied in molten Na-metal batteries [32,33]. In the context of low temperature (<150 °C) molten SMBs, NASICONs exhibit higher ionic conductivities compared to BASE at low temperatures [32]. The poor Na wetting on BASE has been frequently ascribed to the high surface tension of molten Na and the formation of Na_2_O resulting from the reaction between molten Na and adsorbed water on the BASE surface, or to the presence of surface Ca impurities left over from synthesis [5,6,8]. However, few studies have been focused on improving the molten Na|NASICON interface, and it is unclear if such mechanisms and the abovementioned wetting-improving strategies for BASE can also be extended to NSICON solid electrolytes. To this end, Gross et al. greatly reduced interfacial resistance in sodium symmetric cells operated at 110 °C by an in situ formation of tin-based chaperone phases on NZSP surfaces. In combination with a well-designed revolutionary high-voltage NaI-GaCl_3_ catholyte (Figure 1k), the work enabled stable electrochemical cycling in a molten Na-NaI battery, sustaining more than 6000 h (>8 months) and 400 cycles with no discernible change in the voltage profile (Figure 1l) [34].

## 3. All-Solid-State Sodium-Metal Batteries

There have been quite a few interface engineering works so far aiming to enhance the Na|SE interfacial wettability, and most of them have been applied to NZSP-based solid electrolytes and only a few on BASE-based solid electrolytes [35,36,37,38,39], most likely due to the high room-temperature ionic conductivity (~1 mS/cm), ease of synthesis, and good chemical stability of NZSP [9,40]. Table 1 provides a comparison in symmetric-cell electrochemical performance (in terms of testing temperature, interfacial resistance, critical current density or CCD, and lifespan) of all-solid-state sodium-metal batteries modified with various wetting strategies.

The strategies to improve Na wettability on NZSP can be generally classified into two categories: (1) removing the surface passivation layer and (2) introducing sodiophilic wetting agents. The strategy (1) is mainly realized by mechanical polishing and high-temperature annealing designed to remove the NaOH/Na_2_CO_3_ surface contaminants on NZSP, thereby exposing the intrinsic NZSP surface with higher Na affability [15,41]. According to a recent work by Gao et al., by simply annealing NZSP at 450 °C for 2 h, the surface contamination layer can be effectively removed, leaving a clean and Na-deficient surface with much improved Na wettability (from a molten Na|NZSP contact angle of 122.5° to 72.5°) [15]. The strategy (2) involves multiple methods, including the implementation of Na-based composite anodes [42], surface coatings (i.e., Sn, TiO_2_, AlF_3_, PVDF, etc.) [13,14,43,44,45,46,47], and bulk doping of NZSP [48,49,50]. Zhou et al. demonstrated that heating NZSP over 300 °C results in the in situ formation of a thin interfacial interlayer with good molten Na wettability (Figure 2a,b) [41]. As depicted in Figure 2c, benefiting from the superior wetting and intimate interface contact of Na|H-NASICON, the interfacial resistance of Na||H-NASICON||Na is much lower than that of Na||NASICON||Na. As shown in Figure 2d, the Na||H-NASICON||Na cell affords a steady cycle life up to 550 h at 0.15 mA cm^−2^ and, subsequently, 0.25 mA cm^−2^. The presence of stable voltage curves during long-term cycling implies the dendrite formation is suppressed owing to a better wetting of the interlayer.

Lu et al. proposed a trilayer monolithic Ca-doped NZSP pellet via a simple co-pressing method [48]. As illustrated in Figure 3a, a porous|dense|porous trilayer structure was first constructed, and then a SnO_2_ coating was applied by infusing SnCl_4_ solution into the porous layers, followed by annealing at 500 °C in air. The resulting trilayer electrolyte achieved perfect molten infusion enabled by the capillary force and the sodiophilic SnO_2_ coating. As a result, not only was the ingenious 3D porous electrolyte layer endowed with high ionic conductivity but it also reduced interfacial resistance. In light of these merits, the corresponding symmetric cell could be stably operated at room temperature for 600 h with low overpotentials under current densities ranging from 0.1 to 0.3 mA cm^−2^ (Figure 3b). Another common coating material, TiO_2_, has been also applied to improve the NZSP|Na interfacial wettability. Gao et al. fabricated a two-phase composite NZSP (TiO_2_) in which TiO_2_ served as an additive [13]. TiO_2_ is beneficial for the densification process because it contributes to reducing the sintering temperature, thereby preventing grain growth and encouraging a more uniform size distribution. The TiO_2_ additive filled grain boundaries, as well as coated NZSP particle surfaces, improving molten Na wetting (Figure 3c) and mitigating dendrite growth. As a result, the Na_3_V_2_(PO_4_)_3_||NZSP(TiO_2_) ||Na cell delivered an outstanding rate performance and enhanced cycling stability in comparison to the NZSP cell (Figure 3d,e). Recently, elemental metals alloyable with Na, i.e., Sn, were demonstrated to alleviate the Na|NZSP wetting issue. Oh et al. prepared a composite anode consisting of Na and Na_15_Sn_4_ by mixing Sn particles in molten Na [43]. The optical image of the Na_5_Sn (i.e., the optimal weight ratio of Na to Sn is 5:1) on NZSP and the corresponding cross-section SEM image are illustrated in Figure 4a,b. Obviously, the Na_5_Sn composite showed higher wettability on NZSP owing to reduced surface tension and formed an intimate interfacial contact with no presence of pores. Furthermore, the presence of Na_15_Sn_4_ in the Na matrix improves the kinetics of interfacial Na transport, increasing vacancy diffusion and delaying pore formation at the anode-NZSP interface. Consequently, the robust Na_5_Sn-NZSP interface underwent repeated galvanostatic cycling without failure for 300 cycles at 0.5 mA cm^−2^ (Figure 4c).

Though relatively less studied, some interface engineering approaches were also reported to improve the Na|BASE wetting towards all-solid-state Na-metal batteries, i.e., introducing carbon- [35] and Sn-based materials [36,37,38], in which Sn was shown to be particularly effective in improving interfacial contact between Na and BASE. Figure 4d shows two different wetting behaviors of molten Na on bare and Sn-coated BASE at 150 °C. By depositing a thick Sn layer, the improved wettability of molten Na on BASE could be achieved, which is attributed to the decreased roughness and reduced surface tension of Na, and, more importantly, the formation of a Na_15_Sn_4_ alloy layer that is sodiophilic and can provide the necessary ionic/electronic conduction for Na plating/stripping [36].

Consequently, the symmetric cell using the modified BASE shows smaller voltage hysteresis and more stable galvanostatic cycling stability at 0.5 mA cm^−2^ (Figure 4e). In an innovative manner, Lu et al. demonstrated a triple Na_x_MoS_2_-carbon-BASE nanojunction interface strategy to address the Na|BASE contact challenge [37]. With a rationally designed Na_x_MoS_2_-C-BASE triple junction interface, the adhesion between NA|BASE was dramatically improved, enabling high-capacity cycling of all-solid-state Na-S batteries at a low temperature of 80 °C, offering 500 mAh g^−1^ after 50 cycles. (Figure 4f–h). A recent study by Deng et al. reported that an yttria-stabilized zirconia (YSZ)-enhanced BASE (YSZ@BASE) has an extremely low interface impedance of 3.6 Ω cm^2^ with the Na metal anode at 80 °C, together with an extremely high critical current density of ~7.0 mA cm^−2^. Furthermore, their quasi-solid-state Na||YSZ@BASE||NaNi_0.45_Cu_0.05_Mn_0.4_Ti_0.1_O_2_ full cell achieves a high capacity of 110 mAh g^−1^ with a Coulombic efficiency > 99.99% and retains 73% of the cell capacity over 500 cycles at 4 C and 80 °C. However, the work mainly highlights the stable *β*-NaAlO_2_-rich solid-electrolyte interphase and strong YSZ support matrix that played a critical role in suppressing the Na dendrite instead of a wetting improvement [39].

## 4. Conventional Sodium-Metal Batteries

For conventional sodium-metal batteries using liquid electrolytes, we divide the sodiophilicity regulation strategies based on the types of host materials, including carbon-based hosts, alloy-based frameworks, and metal- and MXene-based skeletons. A comprehensive summary for the electrochemical performances of Na metal anodes modified by different wettability-enhancing strategies is provided in Table 2.

### 4.1. Carbon-Based Hosts

Carbon cloths (CCs) are particularly appealing as scaffolds for advanced sodium-metal anodes because of their combination of mechanical toughness, electrical conductivity, low weight, and cost efficiency, which can effectively accommodate the volume deformation and mitigate dendrite growth during the Na plating/stripping procedure. Unfortunately, conventional CC is generally sodiophobic in nature and thus does not allow for the infusion of molten Na, motivating many modification approaches to introduce a desired sodiophilicity. Wang et al. developed a Na-carbon composite electrode with a sodiophilic matrix by employing a pre-heated CC and an infusion route, which allowed a homogeneous Na^+^ deposition via a capillary action [51]. Similar to this strategy, Go et al. also fabricated nanocrevasse-rich Li/Na metal polyacrylonitrile (PAN)-based carbon composites [52], but a step closer comparison with the former was that large-scale Li/Na metal carbon composites can be fabricated with a simple machine (Figure 5a). Aside from the studies mentioned above, it was widely reported that introducing alloy phases, metal oxide nanoparticles, functional groups, or heteroatom dopants could dramatically improve the sodiophilicity of CC, thereby significantly refining the stripping/plating behavior of Na metal anodes [53,54,55,56,57]. Recently, Wang et al. utilized a simple Na/In liquid immersion in a CC scaffold and a subsequent condensation procedure to synthesize a Na/In/C composite (Figure 5b) [53]. The existence of sodiophilic NaIn and Na_2_In phases on the Na/In/C composite electrode interface, as evidenced by experimental investigations and DFT simulations, is favorable to enhancing Na-ion deposition stability. When the Na/In/C anode was paired with the Na_3_V_2_O_2_(PO_4_)_2_F (NaVPOF) cathode, the full cell achieved a high specific capacity of 88.4 mAh g^−1^ with a capacity retention of 87.6% after 800 cycles at 1 C rate. Ye et al. prepared RuO_2_@CC by a simple solution-based method [54]. After infusion of molten Na, the RuO_2_@CC was converted to Na-Ru@CC anodes. The Ru nanoparticles not only played a role in assisting uniform Na^+^ plating on the 3D carbon framework but reduced the local current density by offering high electrical conductivity. The Na-Ru@CC electrodes afforded a stable cycling over 250 h at 1 mA cm^−2^ under a controlled capacity of 1 mAh cm^−2^. Following a similar approach, Xiong et al. developed ultra-stable 3D-sodium-infiltrated Fe_2_O_3_-coated carbon textile anodes and sodium-infiltrated carbon textile (CT)-based anodes [55,56], both of which exhibited excellent machineabilities and stable cycling stabilities. In addition, a Co nanoparticle/N-doped carbon decorator (Co-VG/CC) was designed by Lu et al. via the approach shown in Figure 5c [57]; the N-doped carbon could be simultaneously installed in the composite during the generation of Co nanoparticles, further enhancing the conductivity and sodiophilicity of Co-VG/CC. As depicted in Figure 5d and the inset, the symmetric cell using Na@Co-VG/CC electrodes exhibited a cycle life of 2000 h with a low overpotential of about 18 mV at 1 mA cm^−2^/1 mAh cm^−2^. Furthermore, in situ optical microscopy observations showed that the Na@Co-VG/CC electrode retained a nearly identical shape throughout the plating process, with no visible dendrites identified. When coupled with the Na_3_V_2_(PO_4_)_3_ (NVP) cathode, the full cell showed a significantly improved cycling stability, with a capacity retention of 90.44% after 370 cycles at 0.2 A g^−1^.

Besides the CC-based sodium composite anodes, carbon papers (CP), carbon felts (CF), carbon sheets (CS), and carbon-fiber-based singles have also been adopted as hosts for sodium metal anodes. Similarly, the heteroatom dopants have been demonstrated effective in improving the surface wettability of the substrate, leading to uniform nucleation and deposition of Na metal. Zhao et al. rationally designed Na@CP-NCNTs composite electrodes by infiltrating Na into CP with N-doped carbon nanotubes (NCNTs) (Figure 6a) [58]. The growth of vertical NCNTs could successfully alter the Na wettability of CPs from “Na-phobic” to “Na-philic”. Therefore, the obtained Na@CP-NCNTs presented a homogeneous local current distribution and an intact 3D skeleton structure after repeated cycles. Nitrogen doping achieved by different N precursors (i.e., urea, melamine, ammonia, hexamethylenetetramine, polypyrrole, thiourea, etc.) has also been investigated but for other applications, such as supercapacitors and Li-ion batteries [85,86]. However, the potential influence of specific functional groups on sodiophilicity and their application in SMBs remain to be studied in depth. Later, Wu et al. successfully fabricated a Na-Na_2_S-CTP (CTP: carbonized tissue paper) hybrid anode by filling sulfur-doped carbon networks with molten sodium [59]. The CTP defects generated during the sulfur doping process, as well as Na_2_S created on the surface of CTP, might lower mass transfer resistance and increase surface mobility during the sodium plating procedure. The alloying strategy could also improve the wettability of CF with Li/Na metal. Zhang et al. formed Li/Na-Sn alloys by introducing SnO_2_ through a convenient solution-based method (Figure 6b) [60]. The surface energy between molten Li/Na and CF greatly decreased due to the as-formed Li/Na alloys, which enables even deposition of molten metal on the CF. The resulting Li/Na-Sn alloy layer could provide plenty of electrochemically active sites to drive uniform Li/Na nucleation and prevent dendrite formation. By introducing robust Co_3_O_4_ nanofibers onto a carbon sheet substrate, Zhao et al. constructed a hierarchical 3D Co_3_O_4_-CS scaffold with excellent wettability for alkali metal anode [61]. The 3D CS offers a primary framework with adequate Na nucleation sites, while Co/Na_2_O nanofibers provide physical incarceration of deposited Na and further reallocate the Na^+^ flux on each carbon fiber. As a result, both Li/Na-Co-CS symmetrical cells exhibit glamorous lifespans with low overpotentials even at high current densities, which is due to the largely reduced local current density and minimal volume expansion. Distinctively, Chi et al. directed molten Na into unmodified carbon felt to fabricate Na/C composite electrodes (Figure 6c) [62]. As shown in Figure 6d,e, the Na/C composite showed lower interfacial resistance than the bare Na electrode after 120 cycles at a high current density of 3 mA cm^−2^. More intriguingly, SEM revealed a flat surface and no Na dendrite occurred. It was demonstrated that the 3D carbon felt reduced the current density of the Na/C composite surface anode during electrochemical cycling, thus leading to highly stable cycling and preventing dendritic Na growth. Furthermore, the Na_0.67_Ni_0.33_Mn_0.67_O_2_||Na/C full cell maintained a higher coulombic efficiency of about 99.9% over 200 cycles at a current rate of 1 C.

Another carbon-based material, graphene, is a typical 2D framework that can effectively decrease the local current density and facilitate uniform Na deposition because of its large specific surface area, ultralight weight, and excellent adsorption in organic solvents. Wang et al. developed a processable and moldable Na@r-GO (r-GO: reduced graphene oxide) composite anode using the molten Na infusion method, in which the GO (graphene oxide) is reduced to r-GO by contacting it with molten Na [63]. The as-obtained composite anode could be molded into various shapes and was stick-resistant. As expected, the plating/stripping behavior of the Na@r-GO composite anode was greatly prolonged electrochemically in both ether and carbonate electrolytes, with suppressed dendritic formation. Wu et al. created a robust ultra-light rGa (reduced graphene oxide aerogel) host by a hydrothermal reduction followed by oriented freeze-drying and transformation of rGa into Na@rGa via molten Na infusion (Figure 6f) [64]. Due to the advantages of ultra-light quality, uniform porous structure, and good wettability with both Na and electrolyte, the host endowed the Na@rGa composite anode with a high energy density and an excellent cycling performance in carbonate-based electrolyte without using any additives. Even at a high current density of 5 mA cm^−2^, the Na@rGa composite anode could retain remarkable cycling stability with no dendrites and low hysteresis (50 mV) over 1000 cycles.

In efforts to inhibit the Na dendrite growth and improve the cyclability of Na metal anodes, other uncommon carbon material bodies (biomass, polymers, MOFs, etc.) have been investigated as effective hosts. Luo et al. prepared a Na-wood composite electrode via encapsulating metallic Na into porous channels within an electrically conductive carbonized wood host (Figure 7a) [65]. Most recently, as depicted in Figure 7b, Li et al. fabricated a 3D oxygen-containing carbonized coconut framework (O-CCF) from biomass waste coconut clothing [66]. Based on first principles and molecular dynamics simulations, a lone Na atom could be strongly adsorbed on O-CCF with C-O (2.04 eV) groups (Figure 7c–e). Accordingly, the 3D O-CCF afforded to regulate the Na nucleation behavior and to prevent the Na dendrites growth, achieving an outstanding cyclability in Na-metal batteries. Even under brutal conditions of a high 5 mA cm^−2^ with a fixed capacity of 10 mAh cm^−2^, the O-CCF still realized a high coulombic efficiency of 99.6% over 1000 cycles. Liu et al. constructed a Na@CC skeleton via carbonization of cotton cloth, followed by molten Na infusion [67]. The successful N and O co-doping in CC not only reduced the local current density but also adjusted the uniform Na deposition. Recently, Liang et al. also demonstrated a nitrogen-doped hard carbon (NHC) composite electrode [68]. The N-functional groups modulated the surface chemistry of the hard carbon from sodiophobic to sodophilic, achieving the same effect as mentioned above: homogenizing the local electron distribution and physically preventing the formation of Na dendrites during the Na plating/stripping process. Mubarak et al. developed highly sodiophilic hollow and mesoporous carbon nanofiber (HpCNF) hosts with abundant defects and nitrogen functional groups through coaxial electrospinning (Figure 7f) [69]. Thanks to the uniform and reversible Na plating aided by the resilient fluorine-rich SEI layer, the Na@HpCNF anode in symmetric cell sustained more than 1000 h at 5 mA cm^−2^ and a capacity of 5 mAh cm^−2^. Tao et al. also reported a lignin-derived carbon nanofiber (LCNF) prepared by electrospinning to encapsulate molten Na [18]. The obtained self-sodiophilic LCNF host can evenly anchor the Na deposits and regulate SEI formation and was thus endowed with high cyclability. In generalized research, Zhu et al. discovered metallic sodium could be controllably deposited through main group II metals (Be, Mg, and Ba) owing to their definite solubilities in sodium [70]. In the case of the 3D hierarchical structure (3DHS) with Mg clusters, a superior “sodiophilic” affinity was observed. After Na infusion, their BET analysis showed that Na-3DHS presents a substantially smaller surface area (3.9 m^2^ g^−1^). Because of the abundance of Mg nucleation seeds, the nucleation barriers of sodium were markedly reduced by further homogeneously spreading Mg clusters in a three-dimensional hierarchical framework. Hence, The Na-3DHS exhibited great cycling stability, including a low overpotential of ~51 mV at 0.5 mA cm^−2^ and 1 mAh cm^−2^ after 100 cycles.

### 4.2. Alloy-Based Frameworks

Recently, “self-healing” liquid Na-K alloys at room-temperature have received more attention as promising Na metal anodes because of their ability to suppress dendritic growth while offering fast charge/discharge capability. However, due to high surface tension, the Na-K alloy anode not only prevents itself from spreading across the surface of the liquid electrolyte but also makes it extremely difficult to disseminate among hosts. What is more, the underlying chemistry of interface formation and carbon/Na-K interaction is still puzzling [71,72,73,74,75,76]. Xue et al. prepared a dendrite-free liquid Na-K anode by absorbing it into carbon paper at 420 °C and thoroughly investigated its dual-anode behavior [71]. The deciding factor for whether a liquid Na-K is a Na or K anode is the energy gained by the insertion of K^+^ relative to Na^+^; if the insertion of K^+^ is more stable, K^+^ is preferred over Na^+^, but, if K^+^ cannot be inserted, Na^+^ is preferred. After that, Xue et al. utilized vacuum infiltration to immobilize a Na-K liquid alloy within porous membranes at an ambient temperature (Figure 8a) [72]. The experiment showed that the liquid anode membrane as obtained was compatible with all carbonate electrolytes but not with ether electrolytes. As a result, the Na-K-Al||NaClO_4_||Na_2/3_Ni_1/3_Mn_2/3_O_2_ cell presented an average coulombic efficiency of 99.8% with little capacity fade over 500 cycles at 1 C (Figure 8b). In order to unlock the underlying chemistry involving interface formation and carbon/Na-K interaction, Zhang et al. reported an ultra-stable and high-capacity anode by using an in-situ-formed graphite intercalation compound (GIC) framework [73]. Designed as Na-K anodes, their unprecedented electrochemical properties could be showcased, which were associated with the synergy effect of fast electron and mass transport of the GIC networks, as well as self-healing behavior of the Na-K alloy. Xie et al. also showed a dendrite-free liquid metal anode made of CC@NaK composites [74]. In a similar manner, the SiCl_4_ electrolyte additive was introduced in an ester-based electrolyte to construct a robust organic/inorganic hybrid interface on the Na-K liquid electrode by Wang et al. [75]. The SiCl_4_ electrolyte additives not only serve as a physical barrier preventing highly reactive alloys from generation of Na_2_CO_3_ but also provided the benefit of fast charge-transfer kinetics. As depicted in Figure 8c, dendrite-free electrochemical behavior was enabled by the hybrid interface connected to the hybrid interphase-modified Na-K (NKC-HI) electrode, as well as the inner liquid alloy. Consequently, the NVP/NKC-HI half cell exhibited a high capacity and outstanding rate performance, with a capacity retention of 103.6 mAh g^−1^ after 1000 cycles at 2 C (Figure 8d). Although extensive research indicated that the well-distributed deposits of liquid metal could be obtained owing to its “self-healing” characteristic, the mossy dendrites of Na still appeared suddenly when using somewhat extreme ratios of liquid Na-K. In this regard, Liu et al. demonstrated a dendrite-free NaK@Na anode by utilizing a liquid alloying diffusion mechanism [76]. Based on in situ optical imaging and theoretical prediction, the NaK@Na anode could deposit isotropically while avoiding dendrite formation.

In addition to the Na-K alloys matrix, some other metallic alloy strategies (Na-Sn and Na-Bi) can also be applied as a sodium metal anode. When compared to Na-K alloy matrix with high electronic conductivity (such as CC [75] or CP [71] host) or ionic conductivity (such as NVP host [87]), it is highly desired that the 3D hosts are both efficient in meeting ion and electron transportation pathways to effective uniform Na^+^ deposition and prevent dendrite growth. Zheng et al. fabricated a percolated Na-Sn alloy/Na_2_O framework with dual ion/electron conductive pathways throughout the Na metal, the uniform, dendrite-free Na metal being realized during repeated Na^+^ plating/stripping (Figure 8e) [77]. As shown in Figure 8f, top-view SEM images of NSCA-31 showcased a uniform, continuous Sn distribution within Na metal. DFT calculations revealed the Na-Sn alloy/Na_2_O framework was associated with a high “sodiophilicity” and a low Na^+^ diffusion barrier. The result is that the NSCA-31 symmetric cell maintains excellent stability for up to 550 h under 2.0 mA cm^−2^, with a capacity of 1.0 mAh cm^−2^ (Figure 8g). Afterwards, Cao et al. incorporated SnO_2_@NZSP into bulk Na metal to obtain information about dendritic-free and stable BH-Na anodes [78]. As a devised hybrid anode, the NZSP functioned as fast Na^+^ conduction, aiming to increase the reaction area of the whole electrode volume during plating/stripping of Na^+^. Analogous to the above two cases, Ye et al. also constructed the periodic alternating of electron and ion conductivity in the 3D-Na_3_Bi alloy framework [79]. Furthermore, the 3D-Na_3_Bi alloy framework could effectively alleviate volume expansion, prevent side reactions with electrolyte, and hinder large dendrite formation due to its electron-conductive, ion-conductive, and sodiophilic nature.

### 4.3. Metal and MXene-Based Skeletons

For the purpose of confining molten Na within 3D current collectors so as to achieve a superior wettability and, in turn, even Na^+^ deposition flux, some of the commercial porous metals (Cu foam, Al foam, Ni foam, and so on) have been used [72,80,81]. Wang et al. presented a composite Cu matrix as a stable host for effective impregnation with molten Na (Figure 9a) [80]. The unique surface property of the matrix could direct Na deposition from the scaffold toward the Na reservoirs within the pores, suppressing huge volume fluctuation and mossy/dendritic Na formation upon plating/stripping. As shown in Figure 9b, the cycle stabilities of bare Na, Na@UCF, Na@S-CF, and Na@O-CF electrodes (UCF, S-CF, and O-CF, respectively, referred to untreated, S-treated, and O-treated Cu foams) in symmetrical cells at 0.5 mA cm^−2^ with a capacity of 1 mAh cm^−2^ were compared. It was obvious that the Na@O-CF delivered the smallest average overpotential of less than 50 mV and an outstanding cycle performance over 400 h. Clearly, the Na@O-CF electrode before and after 30 cycles also exhibited lower interfacial impedance values than Na@S-CF and Na@UCF (Figure 9c). Recently, Xia et al. also constructed in situ a supersodiophilic 3D fluffy surface layer on a Cu foam host (SF-Cu-3.6) via a facile and controllable oxidation treatment strategy [81], and the Na/SF-Cu-3.6 composite anode delivered significantly better rate performance than that of a bare Na anode in NaTi_2_(PO_4_)_3_-based full cells.

Recently, because of its abundant surface functional groups, high electronic/ionic conductivity, and high mechanical modulus, MXene, a remarkable two-dimensional (2D) layered material, has been intensively researched to regulate Li/Na deposition and inhibit dendritic growth [82,83,84]. Fang et al. ingeniously designed the Na-Ti_3_C_2_T_x_-CC film composite, in which the Ti_3_C_2_T_x_ functioned as sodiophilicity and fast electron channels, while the CC served as a durable skeleton to enable moldable and processable metal anodes (Figure 10a) [82]. According to first principles calculations and SEM observations, the protective mechanism of Ti_3_C_2_T_x_ was realized by inducing an effective initial nucleation and producing “sheet-like” Na deposition inherited from the MXene architecture. Fang et al. also successfully prepared the 3D Li/Na-Ti_3_C_2_T_x_-rGO films (Figure 10b) [83]. The abundant function groups existing on the Ti_3_C_2_T_x_ surface contributed to good affinity between the Ti_3_C_2_T_x_-rGO membrane and mixed covalent/ionic bonds (Ti-Li/Na, O-Li/Na, and F-Li/Na), leading to uniform electrochemical deposition and preventing dendritic puncture. The resultant Na-Ti_3_C_2_T_x_-rGO electrodes presented a low overpotential of 20 mV after 800 h cycling at 1 mA cm^−2^ and 1 mAh cm^−2^ (Figure 10c). A hybrid rGO/MXene film was further introduced by Wang et al. [84], where the major advancement was that they realized an optimized N/P ratio of 3.8 in Na_3_V_2_(PO_4_)_3_||Na full cells, stretching the feasibility of their anode for practical sodium-metal batteries.

## 5. Concluding Remarks

To summarize, for molten SMBs such as Na-S and Na-MH batteries, a perfect Na wetting on BASE and a promising Na-S battery performance have been already achieved at a temperature as low as 120 °C [20]. As long as the molten Na is used and sufficient wettability is ensured, the molten SMBs should be basically free of dendrite growth. Given the resolved Na wetting issues, for intermediate-temperature Na-MH batteries using BASE separators, seemingly further temperature reduction from the current record of 190 °C is essentially limited by the catholyte, i.e., NaAlCl_4_ with a melting point of 158 °C. Some potential catholytes, such as NaAlCl_3_I with a much lower melting point of ~85 °C, may be considered for running Na-MH batteries at 100-120 °C that are slightly higher than the melting point of Na at 98 °C. Recently, researchers even proposed a room-temperature Na-CuCl_2_ battery by using NaAlCl_4_⋅2SO_2_ electrolyte that is in liquid state at −40 to 70 °C [88]. On the other side, liquid-metal (i.e., Na-biphenyl) anodes have also been reported to pair with BASE and inorganic catholyte in room-temperature Na-S batteries. Similar studies were covered in our recent review on emerging soluble organic redox materials [89]. Although the abovementioned room-temperature batteries are no longer molten SMBs, such catholyte modification and anolyte introduction imply new directions to make current molten Na-S and Na-MH batteries more commercially relevant.

In the context of all-solid-state SMBs using BASE or NZSP solid electrolytes, substantial progress has been made to improve the Na|SE wetting and reduce the interfacial resistance. As also emphasized in a recent review by Thangadurai et al., behind the difficulties in commercializing all-solid-state Li/Na-metal batteries, one of the main reasons recognized is the still low critical current density (CCD) when employing an elemental Li/Na anode [90]. The CCD generally refers to the current density a Li or Na battery can endure through cycling without cell failure due to dendrite growth. Thus far, most wetting strategies are capable of lowering the Na|SE interfacial resistance to a desired extent; however, they still fall short in boosting the CCD to reach 2 mA cm^−2^, which is required for practical applications [9,90]. Therefore, future research in designing robust Na|SE interfaces should not only improve Na wettability for intimate and stable interfacial contact but pay considerable attention to factors, such as stacking pressure, testing temperature, and interfacial compositions and microstructures, that are critical to the CCD. Besides, the Na metal utilization, N/P ratio, and cell-level energy density ought to be considered, an aspect critical yet just starting to receive due attention in all-solid-state batteries [91].

As far as conventional SMBs are concerned, we have shown that a great deal of research was conducted to develop 3D sodiophilic hosts for Na metal anodes, with significant performance achieved. However, it should be noticed that the attention to cell-level energy density, i.e., concerning the host and wetting agent weights, utilization of infused Na, and N/P ratios for full-cell demonstration, is not enough [84,92]. We note that much of the literature does not report cell-level energy densities and cost analyses because the field is still in its infancy compared to Li-ion. Electrochemical performance data under practically relevant conditions, such as high areal capacity and current density, as well as low N/P ratio, are highly recommended when evaluating the feasibility of a Na anode design and assessing progress in the field.

Overall, this review summarizes recent research efforts on the strategies to improve wettability of Na metal anodes in various Na-metal-based energy-storage devices. As discussed above, wetting-based strategies have shown immense potential for modifying Na metal anodes towards practical energy-storage applications. However, simultaneously meeting key requirements concerning practical applications, including low processing cost, high energy density, long cycle life, high energy efficiency, and high-power capability, still calls for much more investigation on both the material and cell levels. In addition to the critical aspects, such as CCD, Na metal utilization, and N/P ratio, in situ and operando studies coupled with DFT and molecular dynamics calculations are advocated for all three types of SMBs in order to probe the underlying mechanisms of Na stripping/plating, especially in the presence of wetting agents, such as alloying elements, i.e., the preferred nucleation sites, the charge distribution, and the Na diffusivity, as well as their dynamics under different current densities. Such knowledge is expected to accelerate the rational designs of Na metal anodes capable of delivering higher CCDs and Na utilization towards safe, fast-charging, and high-energy-density SMBs.

## Figures and Tables

**Figure 1 materials-15-04636-f001:**
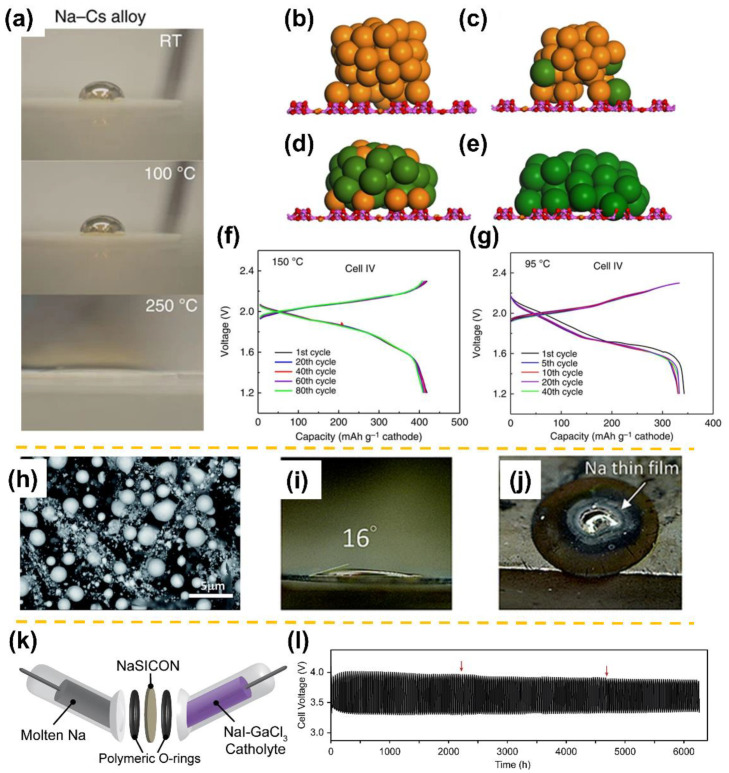
(**a**) Liquid Na-Cs alloy (molar ratio of 1:4) drops on untreated BASE at different temperatures. Simulated atomic structures of liquid Na-Cs alloy droplets on BASE at 100 °C for (**b**) Na, (**c**) Na-Cs (molar ratio of 4:1), (**d**) Na-Cs (molar ratio of 1:4), and (**e**) Cs liquid. The Al, O, Na, and Cs are colored purple, red, orange, and dark green, respectively. Electrochemical performance in terms of voltage profiles of Na-Cs||BASE||S cells operated at (**f**) 150 and (**g**) 90 °C (adapted with permission from Springer Nature). (**h**) SEM image showing the surface morphology of BASE treated with lead acetate trihydrate at 400 °C. The front view (**i**) and corresponding top-view (**j**) of liquid Na sessile drops on the treated BASE measured at 200 °C (adapted with permission from Royal Society of Chemistry). (**k**) Schematic view for the assembly of a Na||NASICON||NaI-GaCl_3_ battery and (**l**) its long-term cycling voltage profile at 5 mA cm^−2^ (adapted with permission from Cell Press).

**Figure 2 materials-15-04636-f002:**
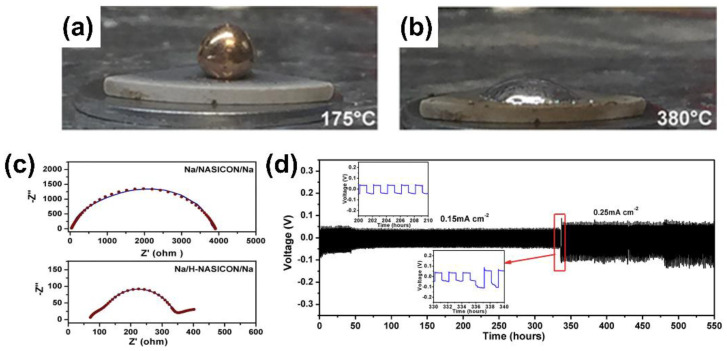
The optical photos of sodium metal on NZSP pellets (**a**) at 175 °C and (**b**) 380 °C. (**c**) The impedance plots of Na||NASICON||Na and Na||H-NASICON||Na symmetric cell at 65 °C. H-NASICON refers to the pellet sample treated at 380 °C. (**d**) Cycling stability of the Na/H-NASICON/Na symmetric cell at 65 °C (adapted with permission from ACS).

**Figure 3 materials-15-04636-f003:**
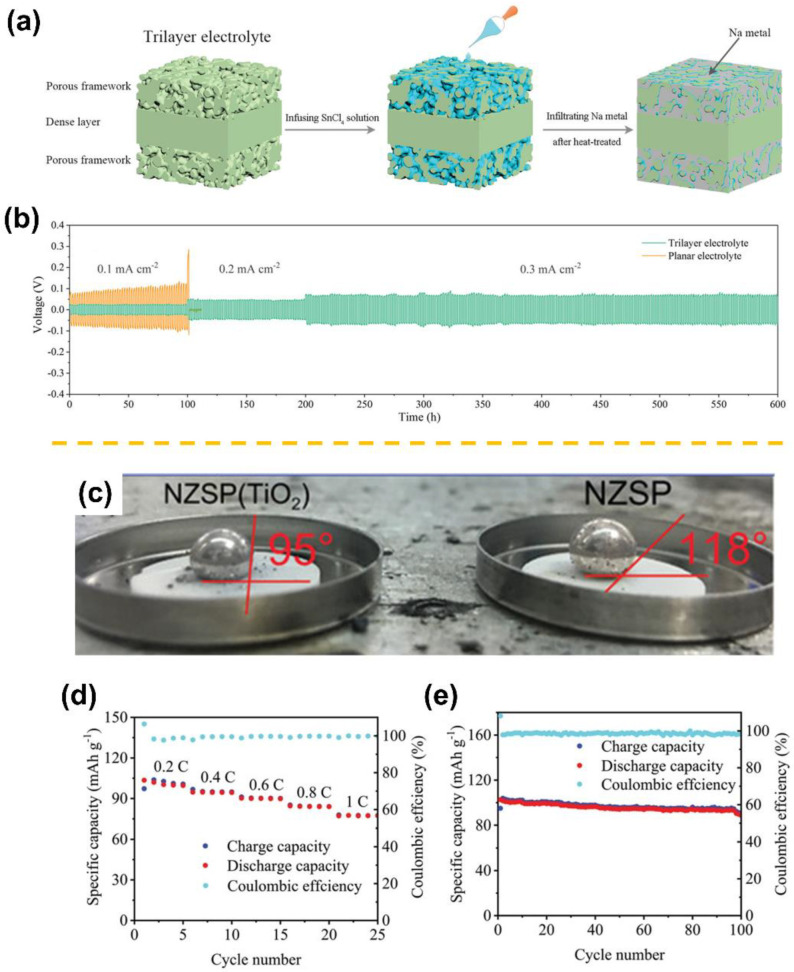
(**a**) Schematic illustration of a trilayer NZSP solid-state electrolyte. (**b**) Voltage profiles of trilayer and planar electrolyte-based symmetric cells. Both electrolytes were surface-modified by SnO_2_ (adapted with permission from Wiley). (**c**) Contact-angle measurements for molten Na on NZSP and NZSP (TiO_2_) surfaces. The Na_3_V_2_(PO_4_)_3_||NZSP(TiO_2_)||Na cell: (**d**) rate performance and (**e**) cycle performance at 0.2 C (adapted with permission from Wiley).

**Figure 4 materials-15-04636-f004:**
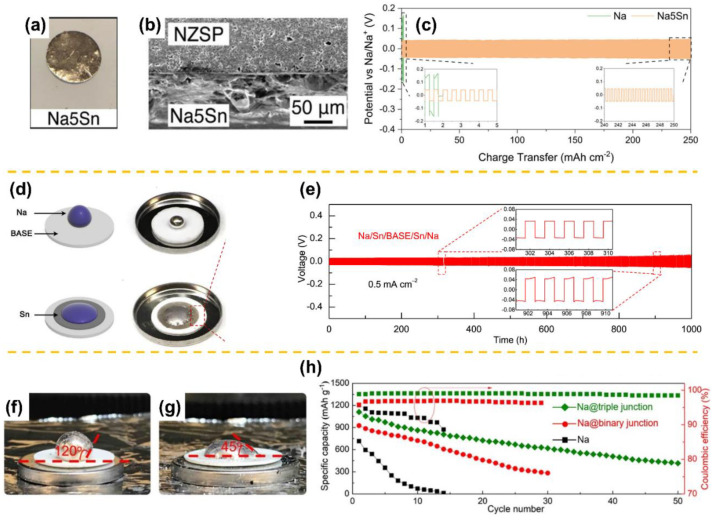
(**a**) The optical images of Na5Sn on NZSP and (**b**) the corresponding cross-section SEM image. (**c**) Cycling stability of Na- and Na5Sn-based symmetric cells at 0.5 mA cm^−2^ (adapted with permission from Wiley). (**d**) Schematic and optical images showing the different wetting behaviors of molten Na on bare and Sn-coated BASE at 150 °C. The sodiophilic phenomenon is observed when Sn serves as an interlayer. (**e**) Galvanostatic voltage profiles of a Na||Sn/BASE/Sn||Na symmetric cell cycled at 0.5 mA cm^−2^. Inset: voltage profiles during 301–311 h and 901–911 h (adapted with permission from Elsevier). Contact angles measured for molten (**f**) Na and (**g**) composite ternary Na-C-Na_x_MoS_2_ anodes on BASE. (**h**) Cycling stability of all-solid-state Na-S cells equipped with different Na anodes at 0.2 mA cm^−2^ and 80 °C (adapted with permission from ACS).

**Figure 5 materials-15-04636-f005:**
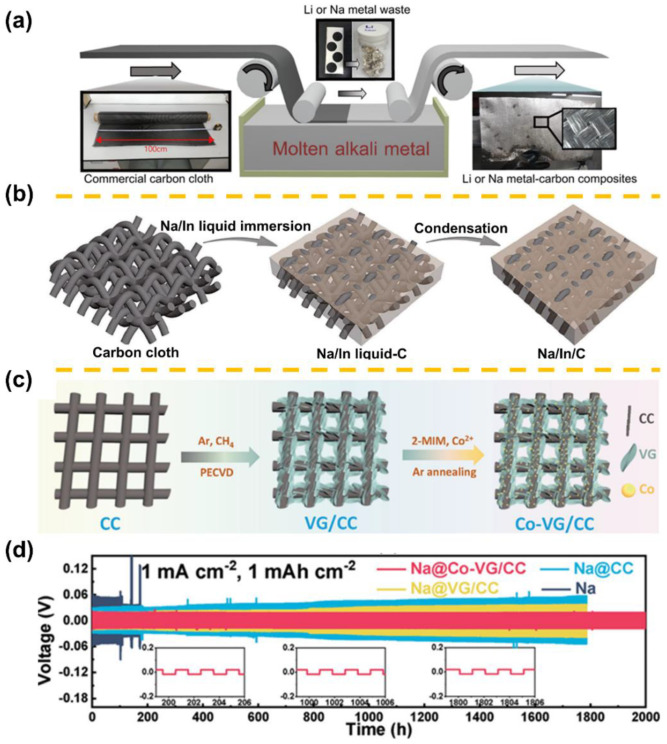
A schematic illustration of (**a**) the fabrication process for scalable Li or Na metal carbon composites production by utilizing a simple machine (adapted with permission from ACS), (**b**) the fabrication procedure of the Na/In/C composite (adapted with permission from Springer Nature), and (**c**) the synthetic process of the Co-VG/CC host. (**d**) Comparison of long-term cyclability of Na@CC, Na@VG/CC, and Na@Co-VG/CC at 1.0 mA cm^−2^ with a fixed capacity of 1.0 mAh cm^−2^. The insets show enlarged profiles of specific cycles (adapted with permission from Wiley).

**Figure 6 materials-15-04636-f006:**
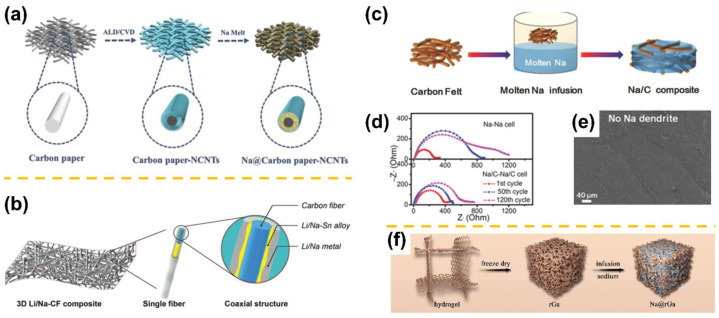
Illustration of the fabrication process of (**a**) the Na@CP-NCNT (adapted with permission from Wiley)*,* (**b**) the Li/Na-CF electrode structure (adapted with permission from Wiley)*,* and (**c**) the Na/C composite anode. (**d**) Nyquist plots of the impedance spectra of the Na/C composite and the bare Na cells after different cycles at 3 mA cm^−2^ current density. (**e**) SEM image of the Na/C composite after 120 cycles at 3 mA cm^−2^ (adapted with permission from Wiley)*.* (**f**) Schematic illustration of the fabrication process of Na@rGa composite anode (adapted with permission from Elsevier).

**Figure 7 materials-15-04636-f007:**
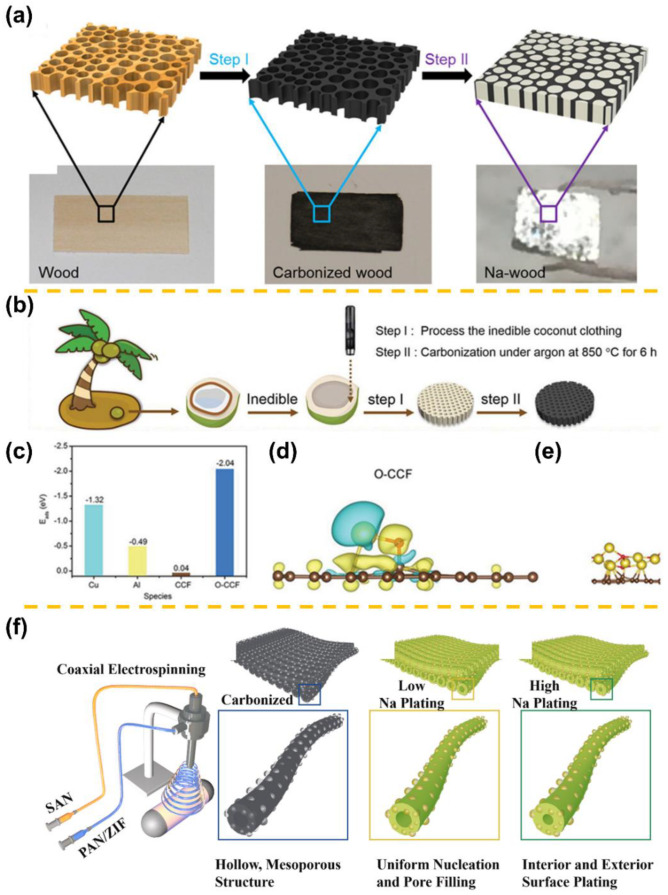
Fabrication process of (**a**) the Na carbonized composite (adapted with permission from ACS) and (**b**) the 3D O-CCF matrix from biomass waste coconut clothing. (**c**) The adsorption energies of Na with Cu, Al, CCF, and O-CCF. (**d**) Differences in charge density of lone Na atoms absorbed on O-CCF. (**e**) The final structures for Na atom adsorption on O-CCF after molecular dynamics simulation (adapted with permission from Wiley). (**f**) Schematic illustration of the coaxial electrospinning setup used for the manufacture of HpCNFs hosts (adapted with permission from Elsevier).

**Figure 8 materials-15-04636-f008:**
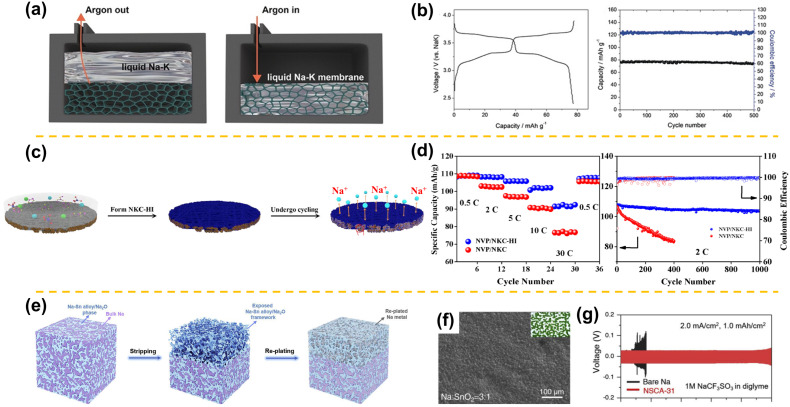
(**a**) Schematic illustration of the fabrication process of a liquid Na-K alloy porous membrane. The porous membrane can be carbon, copper, aluminum, nickel, etc. (**b**) The electrochemical performance of the Na-K-Al||NaClO_4_||Na_2/3_Ni_1/3_Mn_2/3_O_2_ cell at 1 C (adapted with permission from Wiley). (**c**) The schematic showing the Na deposition behavior of the NKC-HI electrode during cycling. (**d**) Rate and long-term cycling performances of NVP||NKC-HI and NVP||NKC half cells (adapted with permission from Elsevier). (**e**) Schematic demonstration of uniform Na^+^ stripping/plating on Na metal within an ion/electron-conductive framework. (**f**) Top-view SEM images of NSCA-31 (Na/SnO_2_ weight ratio of 3:1). (**g**) Cycling performances of bare Na and NSCA-31 symmetric cells at 2.0 mAcm^−2^ with a fixed capacity of 1 mAh cm^−2^ in an ether-based electrolyte (adapted with permission from Elsevier).

**Figure 9 materials-15-04636-f009:**
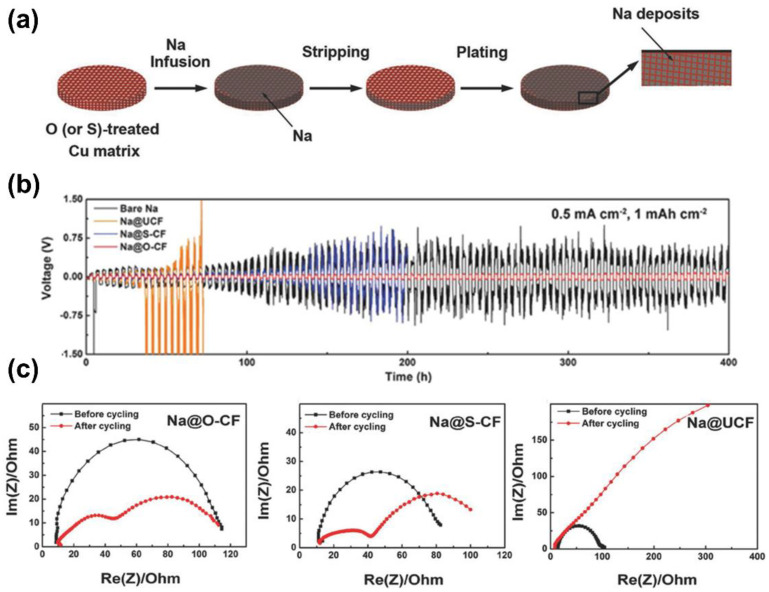
(**a**) Schematic illustration of molten sodium infusion into treated Cu matrix. (**b**) Comparison of the cycling stabilities of bare Na, Na@UCF, Na@S-CF, and Na@O-CF electrodes in symmetrical cells at a current density of 0.5 mA cm^−2^ with a capacity limitation of 1 mAh cm^−2^. (**c**) The Nyquist plots of Na@O-CF, Na@S-CF, and Na@UCF before and after 30 cycles (adapted with permission from Wiley).

**Figure 10 materials-15-04636-f010:**
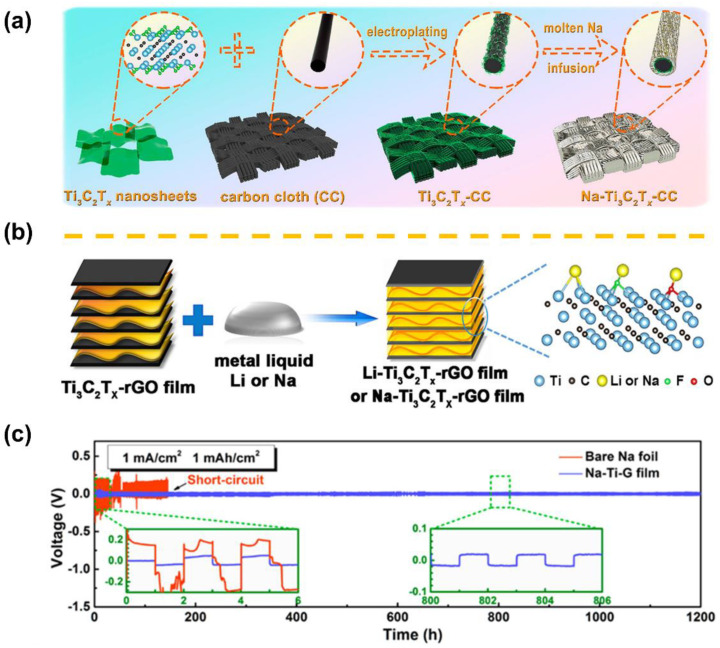
Schematic illustration of the structures of (**a**) Na-Ti_3_C_2_T_x_-CC metal anodes (adapted with permission from ACS) and (**b**) Li/Na-Ti_3_C_2_T_x_-rGO film. (**c**) Comparison of the cycling stabilities of bare Na and Na-Ti-G film electrodes in symmetrical cells at a current density of 1 mA cm^−2^ with a capacity limitation of 1 mAh cm^−2^ (adapted with permission from ACS).

**Table 1 materials-15-04636-t001:** Comparison of symmetric-cell electrochemical performance of all-solid-state sodium-metal batteries modified with various wetting strategies.

Sodiophilicity Regulation Strategy	Testing Temperature	Interfacial Resistance (Ω cm^2^)	CCD (mA cm^−2^)	Lifespan in Symmetric Cells	Ref.
450 °C heat-treated NZSP	25 °C	636	/	1500 h at 0.1 mA cm^−2^; 250 h at 0.3 mA cm^−2^	[15]
Na-NZSP reaction at 380 °C	65 °C	Total resistance: 400 Ω cm^−2^	/	213 h at 0.25 mA cm^−2^	[41]
Na-SiO_2_ composite	25 °C	101	0.5	85 h at 0.2 mA cm^−2^	[42]
NZSP-TiO_2_ composite	25 °C	149	1	750 h at 0.1 mA cm^−2^	[13]
Fluorinated amorphous carbon-regulated NZSP	75 °C	100	/	100 h at 0.5 mA cm^−2^	[14]
Na_5_Sn composite	25 °C	8.5	2.5	500 h at 0.3/0.5 mA cm^−2^	[43]
SnS_2_@NZSP	25 °C	Total resistance: 280	0.9	800 h at 0.1 mA cm^−2^	[44]
SnO_x_/Sn@NZSP	25 °C	3	1	1500 h at 0.1 mA cm^−2^; 500 h at 0.3 mA cm^−2^	[45]
TiO_2_@NZSP	23 °C	101	/	860 h at 0.1 mA cm^−2^	[46]
AlF_3_@NZSP	60 °C	/	1.2	150 h at 0.25 mA cm^−2^	[47]
Porous|dense|porous trilayer NZSP	25 °C	Total resistance: 175	/	400 h at 0.3 mA cm^−2^	[48]
Na_3.4_Zr_1.6_Sc_0.4_Si_2_PO_12_	25 °C	63	/	700 h at 0.1/0.2 mA cm^−2^	[49]
Na_3.4_Zr_1.8_Mg_0.2_PO_12_	25 °C	93	0.95	~5000 h at 0.3 mA cm^−2^	[50]
Disordered carbon tubes @BASE	58 °C	150 Ω cm^−2^	/	1000 h at 0.1 mA cm^−2^	[35]
Sn@BASE	60 °C	9.6 Ω cm^−2^	/	1000 h at 0.5 mA cm^−2^	[36]
Na_x_MoS_2_-carbon-BASE triple junction	80 °C	/	/	200 h at 0.3 mA cm^−2^	[37]
carbon fiber with Sn particles@BASE	25 °C	6.6	1.3	3000 h at 0.2 mA cm^−2^; 400 h at 0.5 mA cm^−2^	[38]
yttria-stabilized zirconia (YSZ)-BASE composite	80 °C	3.6	~7	/	[39]

**Table 2 materials-15-04636-t002:** Comparison of electrochemical performance of Na metal anodes modified by different wettability-enhancing strategies.

Sodiophilicity Regulation Strategy	Electrolyte	Current Density (mA cm^−2^)	Areal Capacity (mAh cm^−2^)	Overpotential (mV)	Cycle Life (h)	Cathode	Full-Cell Electrochemical Performance	Ref.
Na-carbon composite	1 M NaPF_6_ in EC/DMC; 1 M NaPF_6_ in FEC/DMC	1	0.5	80; 50	140; 600	Na_3_V_2_(PO_4_)_3_	➣ 100 mAh g^−1^ and CE of 99.8% with 90% capacity retention after 1000 cycles at 3 C	[51]
Na-C composite	1 M NaCF_3_SO_3_ in DME	1; 3	/	18; 25	200; 80	Seawater	➣ 1106 mAh g^−1^, 94.8% of the theoretical capacity of Na metal (1166 mAh g^−1^)	[52]
Na-In-C composite	1 M NaClO_4_ in EC/PC with 5 wt% FEC	1; 2; 5; 1	1; 1; 1; 5	51; 100; 250; 50	870; 710; 560; 600	Na_3_V_2_O_2_(PO_4_)_2_F	➣ 88.4 mAh g^−1^ with a capacity retention of 87.6% after 800 cycles at 1 C; ➣ Capacity attenuation of 0.019% per cycle	[53]
Na-Ru-cabon cloth	1 M NaClO_4_ in carbonate electrolyte with 5% FEC	1	1	12.5	250	/	/	[54]
Na-Fe_2_O_3_-carbon textile	1 M NaClO_4_ in EC/DMC	1; 3; 5	1; 1; 1	20; 70; 120	333; 222; 139	/	/	[55]
Na-3D SnO_2_ carbon textiles	1 M NaClO_4_ in EC and DMC	1	1	50	222	/	/	[56]
Na-Co nanoparticle-N-doped carbon	1 M NaClO_4_ in EC/DMC/EMC with 2 wt% FEC	1; 3; 5	1; 6; 5	270; 26; 18	280; 1000; 2000	Na_3_V_2_(PO_4_)_3_	➣ Capacity retention of 90.44% after 370 cycles	[57]
Na-carbon paper-N-doped carbon nanotubes	1 M NaPF_6_ in EC/PC	3; 5; 5	1; 1; 3	200; 120; 200	180; 140; 90	/	/	[58]
Na-Na_2_S-carbon composite	1 M NaClO_4_ in EC/DEC with 10% FEC	1; 4	0.5; 2	50; 89	300; 150	Na_3_V_2_(PO_4_)_3_	➣ 54 mAh g^−1^ after 500 cycles at 1 C Capacity retention of about 67.5%	[59]
Na-carbon fiber composite	1 M NaClO_4_ in EC/DMC/EMC with 5% FEC	0.5	1	50	300	/	/	[60]
Na-Co_3_O_4_ nanofiber-carbon sheet	1 M NaClO_4_ in EC/DMC/EMC with 5% FEC	1; 2; 1	1; 1; 3	80; 110; 70	250; 140; 240	Na_3_V_2_(PO_4_)_2_F_3_	➣ 94.3% retention after 100 cycles at 1 C	[61]
Na-carbon felt composite	1 M NaClO_4_ in EC/PC	1; 3; 5	2; 2; 2	20; 50; 100	120; 120; 120	Na_0.67_Ni_0.33_Mn_0.67_O_2_	➣ 72 mAh g^−1^ with a CE of 99.9% after 200 cycles at 1 C	[62]
Na-reduced graphene oxide	1 M NaPF_6_ in EC/PC	0.25; 0.5	0.25; 0.25	90; 110	300; 60	Na_3_V_2_(PO_4_)_3_	➣ Cycling performance slightly enhanced	[63]
Na-reduced graphene oxide aerogel	1 M NaClO_4_ in EC/DEC	0.5; 3; 5	0.5; 2; 1	35; 120; 50	350; 120; 400	Na_0.67_Ni_0.25_Mn_0.75_O_2_	➣ 70 mAh g^−1^ and a CE of 99.8% after 110 cycles at 0.5 C ➣ 79 mAh g^−1^ and CE of 99.5% after 100 cycles at 0.1 C	[64]
Na-carbonized wood	1 M NaClO_4_ in EC/DEC	0.5; 1; 1	0.25; 0.5; 1	30; 62.5; 70	250; 250; 500	/	/	[65]
Na-oxygen-containing carbonized coconut framework	1 M NaPF_6_ in diglyme	10; 30; 50	1; 1; 1	5.3; 12; 22	700; 675; 400	Na_3_V_2_(PO_4_)_3_	➣ Capacity retention of 96% after 100 cycles	[66]
Na-carbon cloth	1 M NaClO_4_ in EC/DEC with 5% FEC	0.3	/	4.8	1600	Na_3_V_2_(PO_4_)_3_	➣ An initial charge capacity of 96 mAh g^−1^ and discharge capacity of 92 mAh g^−1^ ➣ 95.8% CE at first cycle	[67]
Na-N-functionalized hard carbon	1 M NaCF_3_SO_3_ in diglyme	1; 2	1; 2	32; 76	1700; 800	Carbon-coated NaTi_2_(PO_4_)_3_	➣ Capacity retention of 92% after 800 cycles at 1 C	[68]
Na-hollow and mesoporous carbon nanofiber	1 M NaCF_3_SO_3_ in DEG/DME	3; 5	3; 5	40; 50	2400; 1000	Na_3_V_2_(PO_4_)_2_F_3_	➣ 115 mAh cm^−2^ and a CE of 99.4% after 500 cycles at 1 C ➣ 93 mAh cm^−2^ after 200 cycles at 4 C	[69]
Na-3D hierarchical structure	1 M NaClO_4_ in EC/DEC with 5% FEC	0.5; 1	1; 1	27; 60	1350; 380	Na_3_V_2_(PO_4_)_3_	➣ Stable cycling stability of 900 cycles at 10 C	[70]
Liquid Na-K	1 M NaClO_4_ in PC with 10 wt% FEC	/	/	/	/	Na_3_V_2_(PO_4_)_3_	➣ Capacity retention of 85% after 100 cycles	[71]
Na-K-Al	1 M NaClO_4_ in PC with 10 wt% FEC	/	/	/	/	Na_2/3_Ni_1/3_Mn_2/3_O_2_	➣ CE of 99.80% after 500 cycles at 1 C	[72]
Na-K-graphite intercalation compound	1 M NaClO_4_ in 1:1 EC/DEC	0.4	0.4	380	400	Na_2_MnFe(CN)_6_	➣ High-capacity retention after 100 cycles	[73]
NaK-Carbon cloth	1 M NaClO_4_ in PC with 5 wt% FEC	2	2	150	1800	Na_3_V_2_(PO_4_)_3_	➣ 72.76 mAh g^−1^ after 1000 cycles at 10 C	[74]
Na-K treated by 0.1 M SiCl_4_-contained electrolyte	0.8 M NaFSI in EC/DMC	1	1	200	2000	Na_3_V_2_(PO_4_)_3_	➣ 103.6 mAh cm^−2^ and capacity retention of 98% after 1000 cycles at 2 C	[75]
NaK-Na	1 M NaClO_4_ in EC/DMC/EMC with 5 wt% FEC	20; 40	1; 3	25; 25	1200; 2800	Na_3_V_2_(PO_4_)_3_	➣ High-capacity retention rate of 100% after 500 cycles at 10 C and 97% after 120 cycles at 0.5 C	[76]
Na-SnO_2_ weight ratio of 3:1	1 M NaClO_4_ in EC/PC	0.5; 1	1; 1	13; 50	160; 300	Na_3_V_2_(PO_4_)_3_	➣ Capacity retention 91.3% after 100 cycles and 82.7% after 300 cycles	[77]
Bulk hybrid Na-metal	1 M NaPF_6_ in EC/DEC/PC with 5 wt% FEC	1; 1; 3	1; 5; 3	35; 36; 25	750; 700; 150	Na_3_V_2_(PO_4_)_3_/C	➣ 102 mAh g^−1^ with capacity decay of 0.012% per cycle at 5 C	[78]
3D-Na_3_Bi alloy host	1 M NaClO_4_ in EC/DEC with 5 wt% FEC	1	1	80	700	Na_3_V_2_(PO_4_)_3_	➣ Cycling performance slightly enhanced	[79]
Na-oxygen treated Cu foam	1 M NaClO_4_ in EC/DEC	0.5; 1; 2	1; 1; 3	50; 100; 120	400; 200; 300	Na_3_V_2_(PO_4_)_3_	➣ High initial capacity of over 100 mAh g^−1^ ➣ CE almost 100% after 100 cycles at 5 C	[80]
Na-CuO/Cu_2_O surface layer on the Cu foam	1 M NaCF_3_SO_3_ in diglyme	0.5; 0.5; 1	0.5; 2.5; 1	12; 12; 19	1000; 1000; 400	NaTi_2_(PO_4_)_3_	➣ 101 mAh g^−1^ with a capacity retention of 82.8% at 100 C	[81]
Na-Ti_3_C_2_T_x_-carbon cloth	1 M NaClO_4_ in EC/DMC/EMC with 5.0% FEC	5; 3	1; 1	20; 62	300; 300	Na_3_V_2_(PO_4_)_3_	➣ CE of 98% from 50 to 1000 mA·g^−1^ ➣ Stable cycling performance under 0°, 90°, and 270° of bending at 500 mA·g^−1^	[82]
Na-Ti_3_C_2_T_x_-reduced graphene oxide	1 M NaClO_4_ in EC/DMC/EMC with 5.0% FEC	1; 3	1; 1	20; 33	1200; 500	/	/	[83]
Na-reduced graphene oxide/MXene	1 M NaPF_6_ in diglyme	1; 3	1; 1	34; 85	1700; 1600	Na_3_V_2_(PO_4_)_3_	➣ N/P ratio = 3.8:1 ➣ Capacity retention of 88.3% after 280 cycles	[84]

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
