# Peer review of "Stabilizing Metallic Na Anodes via Sodiophilicity Regulation: A Review"

_materials, 2022, doi:10.3390/ma15134636_

Round 1

Reviewer 1 Report

The presented review is devoted to the current challenges and strategies regarding stabilizing sodium-metal anodes in sodium-metal batteries.

The sodium anode is the essential component of energy storage systems, so the topic of the manuscript is valuable for science and practice. This work may be of interest to community that deals with the applications of the batteries in emerging energy storage. In this regard the presented review can inspire new ideas and concepts in resolving critical metallic anode issues and broadening the applications of sodium-metal batteries in individual battery systems.

The scientific soundness of the manuscript is high. It is well organized, well written and readable. The used English is correct.

The work is sufficiently detailed, scientifically documented and conclusive. The total amount of references is 89 (including the most recent articles from 2022), which is enough for a review type of paper. In 15 references, the co-authors of the articles are the co-authors of the currently reviewed manuscript, so the proportion of references is acceptable.  

In my opinion the reviewed manuscript is interesting and worth of publication. I believe it could be published in the "Materials".

I point out only few minor aspects: 

1.     It is commonly accepted that the phrases: "Wu and colleagues", "Mubarak and colleagues", etc., are not used in scientific publications. So, all these phrases should be replaced throughout the manuscript with: "Wu et al.", "Mubarak et al.", etc.

2.     The same spelling of units should be used throughout the manuscript. In the work under review, the authors use the spelling "mA cm−2" in some sections, and "mA/cm2" in other sections. 

3.     Following the editor's instructions for authors, journal references must cite also digital object identifier (DOI) where available.

Author Response

Reviewer: 1

COMMENTS FOR THE AUTHORS

The presented review is devoted to the current challenges and strategies regarding stabilizing sodium-metal anodes in sodium-metal batteries.

The sodium anode is the essential component of energy storage systems, so the topic of the manuscript is valuable for science and practice. This work may be of interest to community that deals with the applications of the batteries in emerging energy storage. In this regard the presented review can inspire new ideas and concepts in resolving critical metallic anode issues and broadening the applications of sodium-metal batteries in individual battery systems.

The scientific soundness of the manuscript is high. It is well organized, well written and readable. The used English is correct.

The work is sufficiently detailed, scientifically documented and conclusive. The total amount of references is 89 (including the most recent articles from 2022), which is enough for a review type of paper. In 15 references, the co-authors of the articles are the co-authors of the currently reviewed manuscript, so the proportion of references is acceptable.  

In my opinion the reviewed manuscript is interesting and worth of publication. I believe it could be published in the "Materials".

Response:

Thanks for your constructive comments and interest in our work!

  1. It is commonly accepted that the phrases: "Wu and colleagues", "Mubarak and colleagues", etc., are not used in scientific publications. So, all these phrases should be replaced throughout the manuscript with: "Wu et al.", "Mubarak et al.", etc.

Response:

Thanks for the valuable suggestion! All the author-name issues have now been modified as suggested wherever applicable.

  1. The same spelling of units should be used throughout the manuscript. In the work under review, the authors use the spelling "mA cm−2" in some sections, and "mA/cm2" in other sections. 

Response:

Thanks for pointing out this issue! All units of mA/cm2 have been changed to "mA cm-2" to be consistent.

  1. Following the editor's instructions for authors, journal references must cite also digital object identifier (DOI) where available.

Response:

Thanks for the reminder! The DOIs of all journal references have now been provided in the Reference section.

Reviewer 2 Report

The manuscript entitled “Emerging strategies in improving Na wetting for progressive Na-metal batteries ". The work is well written and it presents a very good scientific quality. Besides, the topic is really relevant for both materials and energy storage fields. However, when is compared to the previous reports published in the literature, this work lacks some important information and discussion. Before I recommend its acceptance, some points must be clarified and a moderate revision is needed.

Some other issues that need to be addressed are:

1.    I am not sure I agree with the title. It does not look bold. Try to improve it.

2.    In the abstract….” We conclude with a discussion of the overlooked while critical aspects in existing strategies for individual battery system, and propose promising areas that we believe to be most urgent for further pursuit”. Maybe some specific information should be written to give to the readers a less vague overview.

3.    The main problem statement and justification for the research has not been clearly stated.

4.    It is not clear the contribution of the manuscript to the empirical literature.

5.    Would you explicitly specify the novelty of your work? The main novelty in this work must be clearly pointed out in the introduction.

6.    The authors should mention on the concept of this work with the progress against the most recent state-of-the-art similar studies.

7.    I find there is no convincing link between the motivations for doing the paper and the way it has been conducted as well as the conclusions reached.

8.    The limitation of this study needs to be provided as well.

9.    The authors talk a lot about wetting strategies. Anything about hydrophilicity-hydrophobicity was explicated. These characteristics are highly important to improve the wettability of any kind of material. Shouldn’t be interesting to address it in the paper?

10.  What about the grafting of specific functional groups on an anode surface to improve its wettability? Can the authors be bolder in this regard? Literature shows many works in this regard, and heteroatom doping can be a possibility. The 2 references below can help the authors in this regard.

https://doi.org/10.3390/ma11010134

https://doi.org/10.3390/nano11020424

Author Response

Reviewer: 2

COMMENTS FOR THE AUTHORS

The manuscript entitled “Emerging strategies in improving Na wetting for progressive Na-metal batteries ". The work is well written and it presents a very good scientific quality. Besides, the topic is really relevant for both materials and energy storage fields. However, when is compared to the previous reports published in the literature, this work lacks some important information and discussion. Before I recommend its acceptance, some points must be clarified and a moderate revision is needed.

Response:

We appreciate the reviewer’s constructive comments and kind considerations!

  1. I am not sure I agree with the title. It does not look bold. Try to improve it.

Response:

Thanks for pointing out this issue. We agree with the reviewer and have now changed the title to “Stabilizing Metallic Na Anodes via Sodiophilicity Regulation: A Review”.

  1. In the abstract….”We conclude with a discussion of the overlooked while critical aspects in existing strategies for individual battery system, and propose promising areas that we believe to be most urgent for further pursuit”.Maybe some specific information should be written to give to the readers a less vague overview.

Response:

Thanks for the comment! As suggested, we have specified the terms of “critical aspects” and “promising areas” in the revised manuscript on Page 1: “We conclude with a discussion of the overlooked while critical aspects (Na metal utilization, N/P ratio, critical current density, etc.) in existing strategies for individual battery system, and propose promising areas (anolyte incorporation and catholyte modifications for lower-temperature molten SMBs, cell evaluation under practically-relevant current density and areal capacity, etc.) that we believe to be most urgent for further pursuit”.

  1. The main problem statement and justification for the research has not been clearly stated.

Response:

Thanks for pointing out this issue. According to the reviewer’s suggestion, we have reorganized the Introduction section. On Page 1, we first stated that: “In review of progress made in developing various SMBs including molten, all-solid-state, and conventional liquid-electrolyte-based SMBs, it is generally recognized that all SMBs share a similar Na wetting challenge, although to different extents, and wetting-enhancing therapies such as forming alloys have been extended from one system to another.” Following this, we briefly introduced the Na wetting challenges and strategies for each of the three SMB systems in Paragraph 2-4. Then in Paragraph 5 of Page 2, we have particularly added several sentences stating the problem and justification for the present review: “Clearly, the sodiophilicity (of the solid electrolytes or 2D/3D Na hosts) has been widely deemed vital for achieving stable Na metal anodes in either solid- or liquid-electrolyte-based SMBs. So far, there exist already quite a few comprehensive reviews involving Na-metal anodes and their interphases with liquid or solid electrolytes [2-6, 20]. However, these existing reviews either discuss major advancements in all Na-based batteries [2, 3, 20], or only concern progress made in solid-state Na batteries [6] and dendrite issues [4]. To date, a focused review on a variety of strategies centered on sodiophilicity regulation is not yet available for research in SMBs.” 

  1. It is not clear the contribution of the manuscript to the empirical literature.

Response:

Thanks for your valuable comment! As a short review, the manuscript’s basic objective or contribution to empirical literature is to provide an in-time summary of sodiophilicity-regulation strategies and their corresponding electrochemical performances in different types of Na-metal batteries. To be clearer about this, we have now added two tables (Table 1 and Table 2) on Page 5 and 10 explicitly summarizing various wetting strategies in all-solid-state SMBs and conventional SMBs, respectively. We believe this revised manuscript can draw attentions to the critical Na wetting issues, and may inspire new ideas to advance the Na-metal anodes in different types of SMBs.

  1. Would you explicitly specify the novelty of your work? The main novelty in this work must be clearly pointed out in the introduction.

Response:

Thanks for your valuable comment! We have now specified the motivation and novelty of this work on Page 3: “Therefore, the present review for the first time recognizes the universal Na wetting issues crossing molten, all-solid-state, and conventional liquid-electrolyte-based SMBs, and provides an in-time summary of sodiophilicity-regulation strategies and their corresponding electrochemical performances in different types of SMBs.”

  1. The authors should mention on the concept of this work with the progress against the most recent state-of-the-art similar studies.

Response:

Thanks for your valuable comment! As suggested, we have clearly distinguished the context of this review with some other recent state-of-the-art reviews on Page 2: “So far, there exist already quite a few comprehensive reviews involving Na-metal anodes and their interphases with liquid or solid electrolytes [2-6, 20]. However, these existing re-views either discuss major advancements in all Na-based batteries [2, 3, 20], or only concern progress made in solid-state Na batteries [6] and dendrite issues [4]. To date, a focused review on a variety of strategies centered on sodiophilicity regulation is not yet available for research in SMBs.”

  1. I find there is no convincing link between the motivations for doing the paper and the way it has been conducted as well as the conclusions reached.

Response:

We appreciate the reviewer for pointing out this issue. As detailed in our responses above to the reviewer’s Comments 3-6, based on the reviewer’s valuable suggestions, we have now reorganized the Introduction part to clearly state the motivations and structure of this review paper as clarified below.

On Page 1, we first stated that: “In review of progress made in developing various SMBs including molten, all-solid-state, and conventional liquid-electrolyte-based SMBs, it is generally recognized that all SMBs share a similar Na wetting challenge, although to different extents, and wetting-enhancing therapies such as forming alloys have been extended from one system to another.” Following this, we briefly introduced the Na wetting challenges and strategies for each of the three SMB systems in Paragraph 2-4. Then in Paragraph 5 of Page 2, we have particularly added several sentences stating the problem and justification for the present review: “Clearly, the sodiophilicity (of the solid electrolytes or 2D/3D Na hosts) has been widely deemed vital for achieving stable Na metal anodes in either solid- or liquid-electrolyte-based SMBs. So far, there exist already quite a few comprehensive reviews involving Na-metal anodes and their interphases with liquid or solid electrolytes [2-6, 20]. However, these existing reviews either discuss major advancements in all Na-based batteries [2, 3, 20], or only concern progress made in solid-state Na batteries [6] and dendrite issues [4]. To date, a focused review on a variety of strategies centered on sodiophilicity regulation is not yet available for research in SMBs.” 

After pointing out the motivations for writing this review, we clearly specified the novelty of this review: “Therefore, the present review for the first time recognizes the universal Na wetting issues crossing molten, all-solid-state, and conventional liquid-electrolyte-based SMBs, and provides an in-time summary of sodiophilicity-regulation strategies and their corresponding electrochemical performances in different types of SMBs.” Subsequently, the structure of this review is briefly introduced to guide the readers on how this work is organized: “We will begin with briefly discussing these anode interface engineering strategies in the Section 2. In Section 3, we then discuss recent reports on improving the Na anode wettability on two important SEs, BASE and NASICON-type Na3Zr2Si2PO12 (NZSP) solid electrolytes, assessed in all-solid-state SMBs. In Section 4, carbon, alloy, metal and MXene-based hosts will be individually reviewed in detail, highlighting the role of wet-ting-improvement in stabilizing Na metal anode towards dendrite-free long-cycling SMBs using liquid electrolytes…In the last section, we conclude by pointing out some overlooked yet critical aspects in implementation the wetting-based strategies in individual battery system, and suggest promising directions for further pursuit. This review is expected to inspire new ideas and concepts in resolving critical metallic anode issues and broadening the applications of SMBs in emerging energy storage fields.”

To be more consistent in the structure of the manuscript, we have also reorganized the Conclusion part, starting by pointing out some overlooked yet critical aspects in implementation the wetting-based strategies in individual battery system (see Paragraphs 1-3), and ending with a summary in line with the Introduction and Abstract parts: “Overall, this review summarizes recent research efforts on the strategies to improve wettability of Na metal anodes in various Na-metal-based energy-storage devices. As discussed above, wetting-based strategies have shown immense potential for modifying Na metal anodes towards practical energy-storage applications. However, simultaneously meeting key requirements concerning practical applications, including low processing cost, high energy density, long cycle life, high energy efficiency, and high-power capability, still calls for much more investigations on both the material and cell levels. In addition to the critical aspects emphasized above such as CCD, Na metal utilization, and N/P ratio, in-situ and operando studies coupled with DFT and molecular dynamics calculations are advocated for all three types of SMBs, as to probe the underlying mechanisms of Na stripping/plating especially in the presence of wetting agents such as alloying elements, i.e., the preferred nucleation sites, the charge distribution, and the Na diffusivity as well as their dynamics under different current densities. Such knowledge is expected to accelerate the rational designs of Na metal an-odes capable of delivering higher CCDs and Na utilization towards safe, fast-charging and high-energy-density SMBs.”

We believe with all the modifications made above, the logic structure is now much clearer with strengthened correlations among different sections.

  1. The limitation of this study needs to be provided as well.

Response:

Thanks for your suggestion! We have briefly described the limitation of this study in the Introduction part on Page 3: “We would like to note here that although the three SMB systems are all based on Na metal anodes subjected to similar wetting challenges, they differ from each other a lot in terms of battery components (i.e., electrolyte and/or cathode), cell configuration, operating temperature, etc. Such differences limit the comparisons in the wetting strategy and the battery performance to individual SMB system.”

  1. The authors talk a lot about wetting strategies. Anything about hydrophilicity-hydrophobicity was explicated. These characteristics are highly important to improve the wettability of any kind of material. Shouldn’t be interesting to address it in the paper?

Response:

We appreciate your insightful comment! Indeed, sodiophilicity/sodiophobicity are the underlying characteristics governing the wetting properties of Na as the reviewer pointed out. While we agree with the reviewer on this matter, we found, while examining the literature, that (i) the key parameters such as the contact angle was reported at temperatures ranging from 100 to 380 °C, making it difficult for comparison; (ii) so far very few works reported theoretical calculations based on DFT combined with Young-Dupre equation (Gao et al. Chemistry of Materials 2020, 32, 3970; Lu et al. Nature communications 2014, 5, 1, 1) to clarify wetting-enhancement mechanisms (in terms of, i.e., adsorption energies and work of adhesion). These facts make it difficult to construct a quantitative interpretation of the sodiophilicity/sodiophobicity due to the lack of experimental and theoretical data. Therefore, we consider adopting cell performance parameters (Table 1 and Table 2) to evaluate the effectiveness of different wetting strategies more suitable and convenient in the present review.

  1. What about the grafting of specific functional groups on an anode surface to improve its wettability? Can the authors be bolder in this regard? Literature shows many works in this regard, and heteroatom doping can be a possibility. The 2 references below can help the authors in this regard.

https://doi.org/10.3390/ma11010134 https://doi.org/10.3390/nano11020424

Response:

Thank you for your constructive comments. According your advice, by, we have added some sentences about nitrogen heteroatom doping on Page 14: “Nitrogen doping achieved by different N precursors (i.e., urea, melamine, ammonia, hexamethylenetetramine, polypyrrole, thiourea, etc.) have also been investigated but for other applications such as supercapacitors and Li-ion batteries [88, 89]. However, the potential influence of specific functional groups on sodiophilicity and its application in SMBs remain to be studied in depth.” The two references are added into the revised manuscript:

  1. Reis, G. S. D.; Oliveira, H. P.; Larsson, S. H.; Thyrel, M.; Claudio Lima, E., A Short Review on the Electrochemical Performance of Hierarchical and Nitrogen-Doped Activated Biocarbon-Based Electrodes for Supercapacitors. Nanomaterials (Basel) 2021, 11, (2), 424. DOI: 10.3390/nano11020424.
  2. Li, J.; Qian, Y.; Wang, L.; He, X., Nitrogen-Doped Carbon for Red Phosphorous Based Anode Materials for Lithium Ion Batteries. Materials (Basel) 2018, 11, (1), 134. DOI: 10.3390/ma11010134.

Reviewer 3 Report

Chenbo Yuan and co-workers review recent research on one of key characteristics of sodium-metal batteries, wettability of sodium to solid electrolyte. In fact, the authors introduce already published works by focusing on sodium wetting. However, I feel that this manuscript is just overviewing already published works which does not show any systematic interpretation from the viewpoint of reaction mechanism by breaking down to elementary process and clear strategies by the systematic interpretation. Therefore, few additional information to the bundle of papers is included. The authors are advised to put valuable information from overview of already reported works.

Other minor concerns are shown below.

1.       Figures are too small and one figure includes information from different topics. Figures for different experiments should be shown separate figure with different figure number and be enlarged for the readers can read easily without loupe.

2.       Many abbreviations are used without full spelling at the first appearance.

3.       In page 11, the authors claimed that graphene is a typical 3D framework. I disagree this opinion. Graphene is a typical 2D material.

4.       In Figure 5f and corresponding description in the main text. The authors discuss the rGa. What is rGa? If it is reduced gallium, it is not carbon based hosts and should not be shown in this section.

From above reasons, the authors are advised to revise the manuscript.

Author Response

Reviewer: 3

COMMENTS FOR THE AUTHORS

Chenbo Yuan and co-workers review recent research on one of key characteristics of sodium-metal batteries, wettability of sodium to solid electrolyte. In fact, the authors introduce already published works by focusing on sodium wetting. However, I feel that this manuscript is just overviewing already published works which does not show any systematic interpretation from the viewpoint of reaction mechanism by breaking down to elementary process and clear strategies by the systematic interpretation. Therefore, few additional information to the bundle of papers is included. The authors are advised to put valuable information from overview of already reported works.

Response:

Thanks for your valuable suggestions! We agree with the reviewer that a systematic interpretation would be more interesting. Unfortunately, one limitation of this short review is that it is difficult to systematically compare and analyze all wetting strategies, as we have clarified on Page 3 as a  response to the Reviewer 2’s  Comment 9,: “We would like to note here that although the three SMB systems are all based on Na metal anodes subjected to similar wetting challenges, they differ from each other a lot in terms of battery components (i.e., electrolyte and/or cathode), cell configuration, operating temperature, etc. Such differences limit the comparisons in the wetting strategy and the battery performance to individual SMB system.”

Regarding the reaction mechanism, for most wetting strategies, the wetting agents were introduced to alloy with Na thereby improving the sodiophilicity. Wherever the mechanism analysis is present in the literature, the relevant information (i.e., alloy formation of Na15Sn4, Na-Cs, and Na3Bi) is already available and discussed in detail in this review. Besides, we have added a most recent work demonstrating an improved Na wetting on PbOx/C-coated BASE along with the reaction mechanism on Page 4: “In an effort to reduce the toxic Pb amount, the same group developed a coating on BASE consisting of a porous carbon network decorated with PbOx (0≤x≤2) nanoparticles [22]. An improved molten sodium wetting with a spreading temperature as low as 110 °C was achieved along with a ultrastable symmetric cell cycling for 6000 cycles at 120 °C. They found that sodiation could eventually break through the oxide shell and form metallic components at higher temperatures, as shown in Eq. 1

PbO, PbO2 + Na → Pb0 + Na-Pb + byproducts (Na2O)               (1)

Furthermore, this modification drastically lowered the Pb content to approximately 6 wt.% in the anode interface, and a complete elimination of the toxic Pb was also demonstrated by replacing Pb with environmentally benign Sn [22].”

However, it is difficult to provide more systematic mechanism discussions from a viewpoint of intrinsic characteristics such as sodiophilicity/sodiophobicity governing the wetting properties of Na, due to the limited availability of experimental and theoretical data. For one thing, key parameters such as the contact angle was reported at temperatures ranging from 100 to 380 °C in the literature, making it difficult for comparison. Meanwhile, so far so far very few works reported theoretical calculations based on DFT combined with Young-Dupre equation (Gao et al. Chemistry of Materials 2020, 32, 3970; Lu et al. Nature communications 2014, 5, 1, 1) to analyze the underlying mechanisms (i.e., in terms of adsorption energies and work of adhesion) responsible for wetting enhancements.

Considering these facts, adopting cell performance parameters to evaluate the effectiveness of different wetting strategies seems more suitable and convenient in the present reviews. Nonetheless, according to the reviewer’s suggestion, we have tried to extract more valuable information from the literature, and added two table, Table 1 and Table 2, to provide an in-time summary of sodiophilicity-regulation strategies and their corresponding electrochemical performances in different types of SMBs.

  1. Figures are too small and one figure includes information from different topics. Figures for different experiments should be shown separate figure with different figure number and be enlarged for the readers can read easily without loupe.

Response:

Thanks for your detailed and valuable comments! To improve the image readability and avoid possible confusion, we have resized the images and added dashed lines in yellow to separate figures from different literatures. We apologize for not increasing the number of figures, as we want to keep ~8 figures for this short review.

Besides, we have carefully checked all the figures to make sure every figure contains only data belonging to the same topic. To be specific, the molten sodium-metal batteries, all-solid-state sodium-metal batteries, and conventional sodium-metal batteries were represented in Figure 1, Figures 2-3, and Figures 4-8, respectively, without overlapping.

  1. Many abbreviations are used without full spelling at the first appearance.

Response:

Thanks for pointing out this issue. All of the abbreviations have now been given in full spelling at their first appearances.

  1. In page 11, the authors claimed that graphene is a typical 3D framework. I disagree this opinion. Graphene is a typical 2D material.

Response:

We thank the reviewers’ comment and we apologize for making the mistake. This error has now been corrected on Page 15: “… graphene, is a typical 2D framework that can effectively decrease the local current density…”.

  1. In Figure 5f and corresponding description in the main text. The authors discuss the rGa. What is rGa? If it is reduced gallium, it is not carbon-based hosts and should not be shown in this section.

Response:

Thank you very much for noticing this! We have added the full spelling on Page 15: “Wu et al. created a robust ultra-light rGa (reduced graphene oxide aerogel) host by a hydrothermal reduction…”.

Round 2

Reviewer 3 Report

Chenbo Yuan and co-workers review recent research on one of key characteristics of sodium-metal batteries, wettability of sodium to solid electrolyte. In fact, the authors introduce already published works by focusing on sodium wetting. I still feel that this manuscript is just overviewing already published works which does not show any systematic interpretation from the viewpoint of reaction mechanism by breaking down to elementary process and clear strategies by the systematic interpretation. However, at the current stage, this kind of overview might be valuable for much more systematic understanding. Therefore, I think contents are OK, but figures are still unfriendly to readers. The figure size and letters in the figures are too small to read. The authors are advised to improve the figure size and quality for readers to easily read it.

From above reasons, the authors are advised to revise the manuscript. 

Author Response

Responses to Reviewers’ Comments

Reviewers' Comments to Author:

Reviewer: 3

COMMENTS FOR THE AUTHORS

Chenbo Yuan and co-workers review recent research on one of key characteristics of sodium-metal batteries, wettability of sodium to solid electrolyte. In fact, the authors introduce already published works by focusing on sodium wetting. I still feel that this manuscript is just overviewing already published works which does not show any systematic interpretation from the viewpoint of reaction mechanism by breaking down to elementary process and clear strategies by the systematic interpretation. However, at the current stage, this kind of overview might be valuable for much more systematic understanding. Therefore, I think contents are OK, but figures are still unfriendly to readers. The figure size and letters in the figures are too small to read. The authors are advised to improve the figure size and quality for readers to easily read it.

From above reasons, the authors are advised to revise the manuscript. 

Response:

Thanks for your constructive comments and valuable suggestions! To improve the image readability and avoid possible confusion, we have enhanced the resolution of all the figures and enlarged the letters in them. In addition, to make it easier for the readers to read, we have split Figure 2 and Figure 8 into respectively Figures 2-3 and Figures 9-10 in the revised manuscript. All the relevant descriptions in the main text have been updated accordingly. We believe the figure quality is now much improved and friendly to readers.

Thanks again for your kind considerations!